# Genotypic and Phenotypic Structure of the Population of *Phytophthora infestans* in Egypt Revealed the Presence of European Genotypes

**DOI:** 10.3390/jof8050468

**Published:** 2022-04-30

**Authors:** Sherif Mohamed El-Ganainy, Zafar Iqbal, Hossam Mohamed Awad, Muhammad Naeem Sattar, Abdel Mohsen Tohamy, Ahmed O. Abbas, Julie Squires, David E. L. Cooke

**Affiliations:** 1Department of Arid Land Agriculture, College of Agriculture and Food Sciences, King Faisal University, P.O. Box 420, Al-Ahsa 31982, Saudi Arabia; 2Vegetable Diseases Research Department, Plant Pathology Research Institute, ARC, Giza 12619, Egypt; tohamy12@hotmail.com; 3Central Laboratories, King Faisal University, P.O. Box 420, Al-Ahsa 31982, Saudi Arabia; zafar@kfu.edu.sa (Z.I.); mnsattar@kfu.edu.sa (M.N.S.); 4Agriculture Botany Department, Menofia University, Shibin El-Kom 32415, Egypt; hosamawad80@yahoo.com; 5Department of Animal and Fish Production, College of Agricultural and Food Sciences, King Faisal University, P.O. Box 420, Al-Ahsa 31982, Saudi Arabia; aabbas@kfu.edu.sa; 6The James Hutton Institute, Dundee DD2 5DA, UK; julie.squires@hutton.ac.uk

**Keywords:** *Phytophthora infestans*, Egypt, potato late blight disease, effector diversity, simple sequence repeat marker, genotype

## Abstract

Late blight disease of potato and tomato, caused by *Phytophthora infestans*, results in serious losses to Egyptian and global potato and tomato production. To understand the structure and dynamics of the Egyptian population of *P. infestans*, 205 isolates were collected from potato and tomato plants during three growing seasons in 2010–2012. The characterization was achieved by mating-type assay, metalaxyl sensitivity assay, and virulence pattern. Additionally, genotyping of 85 Egyptian isolates and 15 reference UK isolates was performed using 12 highly informative microsatellite (SSR) markers David E. L. Cooke and five effector (RxLR) genes. Mating-type testing showed that 58% (118 of 205) of the isolates belonged to mating type A1, 35% (71 isolates) to mating type A2, and the rest 8% (16 isolates) were self-fertile. The phenotype of metalaxyl response was represented as 45% resistant, 43% sensitive, and 12% as intermediate. Structure analysis grouped the 85 identified genotypes into two main clonal lineages. The first clonal lineage comprised 21 isolates belonging to A2 mating type and 8 self-fertile isolates. This clonal lineage was identified as Blue_13 or EU_13_A2. The second main clonal lineage comprised 55 isolates and was identified as EU_23_A1. A single isolate with a novel SSR genotype that formed a distinct genetic grouping was also identified. The effector sequencing showed good correspondence with the virulence data and highlighted differences in the presence and absence of loci as well as nucleotide polymorphism that affect gene function. This study indicated a changing population of *P. infestans* in Egypt and discusses the findings in the context of late blight management.

## 1. Introduction

Late blight, caused by a hemibiotrophic oomycete pathogen *Phytophthora infestans* (Mont.) de Bary (1876), is one of the most devastating diseases affecting potato (*Solanum tuberosum* L.) and tomato (*Solanum lycopersicum* Mill.) worldwide [1,2]. The disease has a significant socio-economic impact on potato and tomato production, causing an estimated annual loss of USD 6.7 billion to potato and 100% crop loss to tomato in some parts of the world [3]. The first outbreak of late blight was recorded 170 years ago in the USA, and it subsequently spread across continents to Europe and other parts of the world [4]. *P. infestans* is a persistent threat to potato and tomato cultivation worldwide due to its aggressiveness, host adaptability, high mutation rate, and transportation of infected vegetative material [5]. The origin of *P. infestans* is disputed, but it is believed to have originated in either South America [6] or central Mexico [7]. The genotypic and phenotypic characterization of populations of *P. infestans* has defined the role of certain clonal lineages in disseminating the pathogen across potato-growing regions [8,9]. The infamous Great Irish Famine of the 19th century was caused by the FAM-1 lineage [10], which was subsequently displaced by the US-1 lineage during 1946–1955. Both of these lineages, FAM-1 and US-1, were later displaced by more aggressive and distinct clonal lineages that migrated from Mexico to other parts of the world [10]. Such rapid shifts in the population structure of *P. infestans* have been continued with genetically diverse populations.

The algal-like oomycete *P. infestans* is a diploid heterothallic pathogen that requires association between mating types A1 and A2 to produce long-lived oospores [7]. Sexual progeny from the germinated oospores results in the formation of genetically diverse populations. The dominating *P. infestans* populations in Western Europe comprise more aggressive clonal lineages such as 13_A2 and 6_A1 compared to more diverse sexual populations in central Mexico and Nordic countries [11]. Among A2 mating types, 13_A2 is a highly aggressive and the most dominant metalaxyl resistant lineage that was first reported in 2004 in the Netherlands and Germany [12]. Later, during 2006–2009, it appeared in Poland [13] and the United Kingdom [12]. Subsequently, it caused severe outbreaks in potato and tomato crops in China [9] and India by replacing prior populations of US-1 and related genotypes [3]. Similarly, during a 2008–2014 survey, the 13_A2 genotype accounted for the majority of the population in Algeria, followed by the 2_A1 and 23_A1 genotypes [14]. In Africa, the first *P. infestans* epidemic was reported in 1941 in Kenya, and a year later, it was reported in Uganda. It spread across borders in the successive years, into the Congo in the west and Tanzania in the south [15]. *P. infestans* strain 2_A1 (of European origin) was recently found to be dominating in east African agro-ecological regions, including Burundi, Kenya, Rwanda, Tanzania, and Uganda. The introduction and prevalence of the A2 mating type of *P. infestans* in North Africa may be due to high imports of seed potato from European countries and resulted in the introduction of new European strains in Algeria, Egypt, Morocco and Tunisia [14,16], whereas in North Africa, only the A1 mating type has been persistently found [17]. Egypt is one of 15 countries that exported 5.2% of unprocessed potatoes in 2020, worth USD 221.9 million [18]. *P. infestans* was presumed to have been present in Egypt since at least 1941 [19], and A1 and A2 mating types, as well as self-fertile isolates, have previously been identified in Egypt [20,21,22]. Nevertheless, the distribution of various mating types varied from year to year in Egypt [20,21,22,23,24,25].

Understanding the population structure, evolution, recombining abilities and epidemiology of *P. infestans* in a particular region over time helps decision making in integrated pest management (IPM). In addition to its epidemiological impact, sexual recombination is another pivotal factor that contributes to the rate of pathogen adaptation [26]. Monitoring the A1 and A2 mating-type ratios is important to know the potential extent of sexual recombination and thus the risk of long-lived oospores serving as primary inoculum sources [8]. Microsatellite markers are extremely useful for deciphering the population structure, characterizing the multilocus genotypes, genetic mapping, and estimating the evolutionary processes [27]. An efficient one-step multiplex simple sequence repeats (SSR) protocol employing twelve SSR markers for high-throughput screening of *P. infestans* populations throughout the world has been developed [9]. Effectors are proteins that can be seen as the pathogen’s key weapon to defeat the host’s defense mechanisms [28]. Effector genes condition the susceptibility or resistance of hosts with known R-genes and can both facilitate the infection (virulence factors or toxins) of a host or trigger defense responses (avirulence factors). Thus, phenotypic and genotypic characterization of the pathogen strains may help in the development of more specific disease management strategies. The objectives of this study were to (1) understand the distribution of different mating types of *P. infestans* from potato and tomato in Egypt, (ii) characterize the Egyptian population of *P. infestans* using SSR microsatellite markers, (iii) determine metalaxyl sensitivity and virulence profile and (iv) to study the presence and diversity of key effector genes.

## 2. Materials and Methods

### 2.1. Samples Collection and Isolation of P. infestans

Potato and tomato plants exhibiting blight symptoms were collected from 23 locations of 6 governorates in Egypt, i.e., Menofia, Qalubiya, Kafr El-Sheikh, Dakhilia, Beheira and Gharbia, during three growing seasons 2009/2010, 2010/2011 and 2011/2012 (Figure 1A). Samples exhibiting blight disease symptoms were collected from the leaves and stem of potato and tomato, tomato fruits, and packed into pre-labeled plastic bags.

Sporulating lesions on plant tissues were washed with fresh water and placed in an inverted Petri dish containing water agar; leaf tissues were placed abaxial side up. Plates were incubated in a humid chamber at 15–18 °C for one day, or until new sporulation appeared. Sporangia were picked from the top of the sporulation area and purified by repetitive transfer to rye sucrose agar (RSA) medium (60 g organic rye grains, 20 g sucrose and 20 g agar L^−1^) [29] supplemented with ampicillin (100 mg L^−1^), mycostatin (50 mg L^−1^), pentachlorontrogenzene (10 mg L^−1^), rifampicin (10 mg L^−1^) and benomyl (25 mg L^−1^). The purified isolates were identified using morphological and microscopical methods [30]. Pure mycelial cultures were multiplied and maintained on rye slants and kept at 18 °C for further studies.

### 2.2. Determination of Mating Type

To determine the mating types, each *P. infestans* isolate was paired with A1 and A2 reference isolates on pea agar media. The mycelial plugs of the individual test isolates were placed 4 cm apart from each other on the plates, incubated at 18 °C for 14 days in the dark, and finally assessed for the presence or absence of oospores at the contact interface between the paired isolates [31]. The isolates paired with either the A1 or A2 reference isolate were named accordingly.

### 2.3. Metalaxyl Sensitivity Assay

Radial growth on metalaxyl amended rye agar was used to estimate the metalaxyl sensitivity of 205 purified *P. infestans* isolates as described previously [32,33]. Briefly, ~5 mm mycelial plug from the 14-day-old colony was placed on rye agar plates supplemented with different concentrations of metalaxyl (0, 5 or 100 μg mL^−1^). Control plates were amended with the same concentration of dimethyl sulfoxide (DMSO) as the metalaxyl amended plates. The metalaxyl amended plates were replicated twice and incubated at 18 °C in the dark and the growth response of each isolate was calculated using a standard procedure [34,35]. The isolates growing >40% at both 5 μg mL^−1^ and 100 μg mL^−1^ metalaxyl concentrations were rated as resistant; those sporulating >40% at only a 5 μg mL^−1^ metalaxyl concentration were rated as intermediate; and those sporulating <40% at both 5 μg mL^−1^ and 100 μg mL^−1^ metalaxyl concentrations were rated as sensitive isolates.

### 2.4. Determination of Virulence Profile

The virulence profile of two isolates (EG-75 and EG-96) was tested by inoculating a susceptible potato (*Solanum demissum* cv. Craig’s Royal) and Black’s differential set of potato plants harboring a single resistance gene from R1 to R11 [36] (Appendix A). Potato plants were grown in a 14/10 h day/night cycle at 20 ± 3 °C in the greenhouse at James Hutton Institute, United Kingdom, along with cv. Craig’s Royal (with no known R gene) as a positive control. To identify the physiological race, after 60 days, both plants and detached leaf assay were performed and three leaflets from the middle part of the plants were inoculated. Detached leaflets were placed abaxial side up in a moist plastic tray and then inoculated with 20 μL of sporangial suspension (ca. 150 × 10^3^ sporangia mL^−1^) of each isolate. Trays were covered with plastic bags and placed at 18 °C under a photoperiod of a 14 h light/10 h dark cycle. Seven days post inoculation, isolates were rated as virulent or avirulent based on the lesion development and sporulation as described earlier [37].

### 2.5. DNA Extraction and SSR Marker Analysis

Total genomic DNA was extracted from 85 purified isolates along with the 15 reference isolates [38]. The extracted genomic DNAs were electrophoresed in 1% agarose gel to assess their quantity and quality. Subsequently, the DNA samples were subjected to a 12-plex PCR using 12 simple sequence repeat (SSR) microsatellite markers (D13, G11, Pi4B, Pi04, Pi63, Pi70, PinfSSR2, PinfSSR3 (also referred to as Pi02), PinfSSR4, PinfSSR6, PinfSSR8, and PinfSSR11) (Appendix A) as previously described [9]. Each microsatellite marker was labeled with one of the four fluorescent dyes FAM, VIC, NED, or PET, depending on the expected size of each SSR product to avoid any overlapping allele size ranges. PCR reactions were prepared by multiplexing all the 24 primers in one PCR tube for each DNA sample using the QIAGEN Multiplex PCR Kit (Qiagen, Hilden, Germany). Each single 12.5 µL PCR comprised 1 µL of the DNA extract, 6.25 of 2× Qiagen Type-IT Multiplex PCR mix, 0.44 µL of primer master mix and 4.81 µL of nuclease-free water. Amplification was conducted in a thermocycler (MWG-Biotech, Ebersberg, Germany), with initial denaturation at 95 °C for 5 min, then by 28 cycles of 95 °C for 30 s, 58 °C for 90 s, and 72 °C for 20 s, and a final extension at 60 °C for 30 min.

Electrophoresis and visualization of SSR products were performed on an ABI3730 DNA analyzer (Applied Biosystems, Foster City, CA, USA), following the manufacturer’s instructions. GeneMapper v3.7 (Applied Biosystems) was used to perform the allele sizing and scoring [9].

### 2.6. Data Analysis

Genetic distances and principal component analysis (PCA) were calculated from Bruvo distances [39] using POLYSAT [40] implemented in R (R core team, 2020). For phylogenetic analysis, the distance matrix was exported from POLYSAT to PHYLIP, and the neighbor-joining approach was used to infer the phylogenetic tree [41].

### 2.7. Screening of Effector Genes and Detection of Mutation

The presence and absence of five effector genes, including Avr1, Avr2, Avr2-like, Avr3a, and Avr4, responsible for virulence/avirulence activity, were screened in 18 isolates (Table 1). DNA extraction and PCR assay were performed as described by [12,42]. The PCR amplicons were resolved on 1.8% agarose gel and visualized under the UV gel doc system. Additionally, PCR amplicons were sequenced (ABI3730 automated sequencer, Applied Biosystems, Dundee, UK) to detect nucleotide mutations using the same set of primers used in PCR.

To determine any mutation in the effector genes, the sequences of each effector gene were aligned with the respective genes of the reference isolates, retrieved from the GenBank database (http://www.ncbi.nlm.nih.gov, accessed on 2 February 2022). The MegaX program was used for DNA and amino acid sequence alignments for each of the five effector genes.

## 3. Results

### 3.1. Samples Collection and Isolation of P. infestans

A total of 205 *P. infestans* isolates were obtained during three growing seasons: 2009/2010, 2010/2011, and 2011/2012. The majority of the isolates (91) were obtained during the 2011/2012 growing season, 61 isolates during the 2010/2011 season, and the fewest isolates (53) were obtained during the 2009/2010 growing season (Figure 1B). The majority of *P. infestans* isolates (140 of total 205 isolates) were from the Menofia governorate, followed by the Beheira governorate (32 of total 205 isolates), and then the Kafr El-Sheikh and Qalubiya governorates with five and seven isolates, respectively. The Dakhliya governorate had the fewest, with only two isolates. About 96% of the isolates were obtained from potato crops.

### 3.2. Determination of Mating Type

Out of the collected 205 *P. infestans* isolates, 118 isolates (57.56%) were the A1 mating type, 71 isolates (34.63%) were the A2 mating type, and 16 isolates (7.80%) were self-fertile (SF) in the Egyptian population during study seasons 2009/2010, 2010/2011 and 2011/2012. The mating-type testing data of the 2009/2010 season (53 isolates) showed that 70% of isolates belonged to mating type A1, 17.5% to mating type A2 and 12.5% were SF. Self-fertile isolates were detected in all governorates except Qalubiya (Figure 2A). During the 2010/2011 growing season, 61 isolates were divided into three groups including 39.34% mating type A1, 57.38% mating type A2 and 3.28% were SF. In 2011/2012, of the 91 collected isolates, 62.63% were A1 mating type, 29.67% were A2 mating type and 7.69% were SF. Overall, the occurrence of A1 and SF mating types was lowest in samples from the 2010/2011 growing season (Figure 2B, Table 2).

### 3.3. Metalaxyl Sensitivity Assay

Of the 205 isolates, 92 (45%) were resistant, 88 (43%) were sensitive, and 25 (12%) were of an intermediate type (Table 2). The percentage of metalaxyl resistance and sensitivity of *P. infestans* isolates remained nearly identical over the three years of sampling. During the growing season 2009/2010, comparable proportions (47%) of resistant and sensitive isolates were identified, whereas only three isolates (6%) were intermediate types. However, during the growing season of 2010/2011, the resistant, sensitive and intermediate phenotypes were 52, 36 and 11%, respectively. During the growing season 2011/2012, 35 isolates (38%) were resistant, 41 isolates (45%) were sensitive, and 15 isolates (16%) were intermediate.

### 3.4. Determination of Virulence Profile

The virulence profile of two different isolates of *P. infestans,* EG-96 (genotype 13_A2_84) and EG-75 (genotype 23_A1_18) with dominant genotypes in Egypt, was investigated using differential potato lines and Craig’s Royal as a positive control (Figure 3). In the detached leaflet assay, isolate EG-96 was virulent in all the differential genotypes of potato, including the Craig’s Royal (Figure 3A), whereas in the whole-plant assay, the virulence spectrum was 1, 2, 3, 4, 5, 6, 7, 10, and 11. This indicated that the resistance genes R8 and R9 triggered immunity in the whole plant but were not functional in the detached leaflets. Isolate EG-75 had a virulence profile of 1, 3, 4, 7, 10, and 11 in detached-leaflet assay (Figure 3B), whereas in the whole plant, it was 1, 3, 4, 7, and 11. R10 only triggered immunity in the whole plant assay but it did not recognize the pathogen invasion in detached leaflets.

### 3.5. Genotyping of the Egyptian Population

Using a 12-plex PCR, 85 Egyptian isolates and 15 UK reference isolates were genotyped. In total, 42 allele sizes were detected across the 12 microsatellite SSR loci (Table 3). The G11 locus revealed the most alleles (9), followed by the locus D13 and locus Pi02 with eight and four alleles, respectively. On the basis of the SSR data set, the Egyptian isolates of *P. infestans* were classified into two main clonal lineages.

To infer the lineage of *P. infestans* isolates, we compared the SSR profiles of the Egyptian isolates to a set of reference strains from UK populations with a total of 100 isolates (Figure 4). UK reference isolates are shown with green labels. The 13_A2 genotype formed a single cluster with three sub-clonal variants identified amongst isolates from Egypt (Figure 4A). Two variants at the Pi02 locus were observed; 13_A2-43 and 13_A2-5 sub-clones were the same with alleles 266-268 at this locus, while the third sub-clone, 13_A2-84, contained a new allele combination at this locus (266-272). Three variants were detected at the D13 locus with bi- and tri-allelic 13_A2-43 (136-156 bp), 13_A2-5 (136-154 bp) and 13_A2-84 (134-136-154 bp). Regarding the G11 locus, two variants were detected (13_A2-43, and 13_A2-5) that contained three alleles 154, 160, and 164 bp, whereas 13_A2-84 contained three allele combinations: 154, 156, and 160 bp. No polymorphisms were detected within the three distinct sub-clones in the 13_A2 clonal lineage at eight loci Pi4B, Pi04, Pi63, Pi70, SSR11, SSR4, SSR6, and SSR8. The second cluster contained all isolates that belong to clonal lineage 23_A1 (highlighted with the red line in Figure 4A). Eleven sub-clonal variants were observed within this lineage. Variation was detected in six loci. Tri-allelic isolates were observed at loci Pi02 and G11. The genotypes 23_A1-4, 23_A1-10, 23_A1-12, 23_A1-13, and 23_A1-14 contained three alleles 266, 268, and 270 at the Pi02 locus, while two alleles, 268 and 270 bp, were detected at the locus with genotypes 23_A1-17, 23_A1-18, and 23_A1-19. The second ploidy-level variance was detected at the G11 locus: 23_A1-4 was tri allelic 142, 156, and 206 bp at this locus, 23_A1-10 was 142, 156, and 208 bp, the three genotypes 23_A1-12, 23_A1-13, and 23_A1-14 were 142, 156 and 210 bp at this locus. Conversely, the genotypes 23_A1-15, 23_A1-16, 23_A1-17, 23_A1-118 were bi-allelic at the G11 locus with allele combinations 156 and 210, while 23_A1-19 was bi-allelic at the G11 locus with allele combinations 142 and 206. At the Pi4B locus, two minor variants were detected within the 23_A1 lineage, and the 23_A1-4, 23_A1-10, 23_A1-12, 23_A1-13, 23_A1-14, 23_A1-15, 23_A1-17 and 23_A1-18 genotypes were 213 and 217, while 23_A1-16 and 23_A1-19 were homozygous 213 bp alleles. The locus D13 was the most polymorphic locus in this lineage, and 5 minor variants were detected; the first one was 136 and 210 bp alleles in 23_A1-4, 23_A1-10, 23_A1-12, and 23_A1-19 genotypes, the second one was homozygous 136 alleles in 23_A1-13 genotype, the third one was 142 and 210 bp alleles in 23_A1-14 genotype, the fourth one was 136 and214 bp alleles in 23_A1-15, 23_A1-17, and 23_A1-18 genotypes, and the fifth one was 140 and 214 bp alleles in 23_A1-16 genotype. No variants were detected at the four loci SSR 11, SSR4, SSR6, and SSR8 within this 23_A1 lineage. Two variants were detected at the locus SSR2 in which both a heterozygous 173/175 bp allele combination and homozygous 173 bp allele were detected. Loci Pi70 and SSR8 were identical in both clusters.

In addition, an unrelated isolate (EG-6) was detected with three alleles at loci Pi63 and SSR6 (Figure 4A). The D13 118 bp allele and the SSR6 242 bp allele noted in this isolate termed misc. (for miscellaneous) or ‘other’ were not detected in the two Egyptian dominant lineages. SSR data with isolates details have been uploaded to EuroBlight database: https://agro.au.dk/forskning/internationale-platforme/euroblight/pathogen-monitoring/genotype-map (accessed on 3 March 2022). (Appendix A).

The principal component analysis (PCA) exported from POLYSAT (Figure 4B provided an overview of the variance between the genotyped isolates, and clearly differentiate the two dominant clonal lineages. It also indicates the minor variation within each lineage and that they are unrelated to other reference lineages from the UK population.

### 3.6. Egyptian Population Structure

The characterization of *P. infestans* isolates over time revealed that 13_A2-5 and its minor variant, 13_A2-43, were only present during the 2009/2010 and 2010/2011 seasons and that a new sub-clone 13_A2-84 emerged and was more prevalent in the samples from the 2011/2012 season. (Figure 5). Likewise, three genotypes 23_A1-4, 23_A1-16 and 23_A1-19 were identified during 2010 but not during 2012. Conversely, 23_A1-10, 23_A1-13, 23_A1-14, 23_A1-18, and 23_A1-19 were detected only in 2012. Notably, the population structure of the 23_A1-15 variant remained steady and was detected and predominant in both 2009/2010 and 2011/2012 seasons. These findings based on these samples suggest that the population structure of *P. infestans* in Egypt is dynamic.

### 3.7. Analysis of Effector Genes

The PCR assay to detect five effector genes showed variation either in their presence or absence and, if present, sequencing identified sequence polymorphism amongst the samples of the 13_A2, 23_A1 and ‘misc.’ lineages analyzed (Table 4). The Avr1 gene was not amplified from any of the tested 13_A2 blue lineage or the ‘misc’ isolate but it was amplified from four of the six 23_A1 isolates and from the reference positive control (T30-4). Avr2 was only amplified from one of the eight tested 13_A2 samples (EG-96) but sequencing showed it was a mis-amplification of the Avr2-like gene. However, Avr2 was amplified from all of the six 23_A1 and misc isolates tested. In contrast, Avr2-like was amplified from all 15 Egyptian isolates tested but failed to amplify from the T30-4 control isolate. Both Avr3a and Avr4 were amplified from all nine tested isolates.

Avirulence gene PCR amplicons were sequenced to explore the presence of single nucleotide polymorphisms (SNPs) among the five genes. The obtained sequences were deposited in GenBank, and their accession numbers are presented (Table 5). A summary of the mutation in the sequences of effector genes (avr1, avr2, avr2-like, avr3a and avr4) of the isolates are shown in Figure 6 and Table 6. The sequence data revealed that avr1 had the fewest and avr2-like the most SNPs in these isolates.

SNPs at three nucleotide positions, 425, 441, and 461, were detected in the avr1 effector gene compared to the reference T30-4 isolate. Of these SNPs, two SNPs at 425 and 461 were non-synonymous (N-Syn) and led to a change from isoleucine (I) to threonine (T), and valine (V) to alanine (A), respectively, with the other SNP being synonymous (Syn) type. All these three identified SNPs were located at the C-terminal end of avr1 (Figure 6A). The ‘misc’ and 23_A1 isolate amplified the predicted avr2 gene product, and no SNPs were detected amongst these five samples (Figure 6B).

The avr2-like gene contained the same RLLR-EER motif as the Avr3a and avr2 genes. Two SNPs were detected within the sequences of the avr2-like gene in the six tested isolates. All 13_A2 blue lineage and ‘misc’ isolates contained heterozygous Y (Y = C or T Pyrimidine) and R (R = A or G, Purine) SNPs at 29 and 274 nucleotide positions, respectively. While 23_A1 isolates contained homozygous SNPs at these two positions. Note that the avr2-like locus is absent in T30-4. The avr2-like consists of 166 AA and the two SNPs encode AA changes M^10^ = methionine to T^10^ = threonine and I^92^ = isoleucine to V^92^ = valine (Figure 6D).

In the avr3a gene, two SNPs were detected within the five isolates resulting in the non-synonymous change of two amino acids at the C-terminal end of the Avr3a effector protein. Most isolates contained the homozygous virulent form E^80^ and M^103^ (E^80^ = Glutamic at the 80 position in the C-terminal of Avr3a protein; M^103^ = Methionine at the 103 position in the C-terminal of Avr3a protein). However, one 23_A1 isolate (EG-40) contained the heterozygous E^80^M^103^/K^80^I^103^ virulent form (K^80^ = lysine at the 80 position in the C-terminal of Avr3a protein; I^103^ = isoleucine at the 103 position in the C-terminal of Avr3a protein) due to heterozygous SNPs R (R = A or G, Purine) at the position 238 and K at the position 309 on the Avr3a effector gene (Figure 6C).

In the avr4 gene, 14 SNPs were detected, four of which led to non-synonymous amino acid changes. Three detected SNPs were heterozygous at the 180, 376, and 439 positions. The SNP at the 376 position was a nonsense mutation that led to the introduction of a premature stop codon in the EG-6 (misc) isolate. The third heterozygous non-Syn SNP R (R = A or G) was located at the 439 position, which encodes for E^147^/K^147^ (E = glutamic acid; K = lysine). The remaining ten mutations were the non-Syn type; three of these mutations were before the RFLR domain at the N-terminal, while the rest of the mutations were at the C-terminal. Both forms of AVR4 were virulent in potato clones with Avr4 protein (Figure 6E).

## 4. Discussion

This study of the population of *P. infestans* in Egypt over three consecutive cropping seasons has identified two dominant clonal lineages of European origin (13_A2 and 23_A1). Their phenotypic (virulence and mating type) and genotypic (SSR and effector diversity) traits have been examined and are discussed in relation to other studies.

Isolates belonging to both the A1 and A2 genotypes were detected in all six governorates of Egypt. However, self-fertile isolates were detected in five governorates: Menofia, Beheira, Gharbia, Kafr El-Sheikh and Dakhilia. The proportion of the A1 mating type in this study ranged from 38–70%, indicating a dynamic population; however, no clear trend was apparent over the three seasons examined. The establishment of A1 mating type in the Egyptian population corroborated previous findings that the A1 mating type has not only established itself in the Egyptian environment [45], but it also exhibits high divergence and polymorphism [46]. Another study is in line with our findings found that 15% of 162 isolates of P. infestans sampled during 2005–2006 from Beheira governorate and the surrounding area were A2, 84% were A1, and only two isolates were self-fertile {El-Korany, 2008 #12620]. Samples collected in 2015 to 2016 identified 67 A1, 12 A2 and one self-fertile mating type from Egypt [46]. We examined the infected plant tissues for the presence of oospores, but none were seen in samples from which self-fertile isolates were recovered. However, a low number of oospores were noted in a single blighted stem of cv. Diamont in Tamalay, Menofia Governorate in January 2011 at the end of the growing winter season. Oospores of *P. infestans* have been reported in field-grown potato and tomato in several countries [47,48,49,50]. This indicated that there might be other factors affecting oospore formation in situations where only one mating type is reported under natural infection in fields in Egypt. Investigating such factors may open up a plethora of resources for combating the disease and could be the subject of future study. *P. infestans* is heterothallic and requires two mating types, A1 and A2, for sexual reproduction [51]. However, the term heterothallic is not absolute when applied to this pathogen. There have been many reports of oospores in single isolate cultures of *P. infestans* [4,52,53,54].

The current survey identified a dynamic population amongst isolates of *P. infestans* sampled from Egypt. The cluster analysis (Figure 4B) grouped all the isolates into two main lineages. The first one contained twenty-nine 13_A2 blue clonal lineage isolates. The 13_A2 lineage was first sampled in the Netherlands in 2004, and isolates with this genotype are highly aggressive, spread rapidly and now dominate *P. infestans* populations in large parts of Europe [12]. Some studies show that isolates of the 13_A2 genotype had a shorter latent period, were more aggressive and are able to overcome resistance to some potato cultivars [55]. It was therefore considered a new threat to potato production, and its spread was intensively monitored in Europe (www.euroblight.net, accessed on 2 February 2022). The lineage blue 13_A2 has spread to other potato- and tomato-growing areas of the world and has caused severe outbreaks of late blight on tomato in south-west India [3]; blue 13 lineages have been predominantly found in southwestern China [9,56]. Interestingly, the current study is the earliest date that 13_A2 has been reported in Egypt.

In this study, three distinct sub-clonal lineages of 13_ A2 were detected. This is comparable with the levels of diversity recently reported in Turkey [57], but lower that the twenty sub-clonal types reported in Cyprus [58]. The forms 13_A2-5 and 13_A2-43, previously reported from Europe, have been present in Egypt at least since 2009 [12,56]. The third sub-clonal lineage was 13_A2-84, which differed at three SSR loci and contained a new allele combination (266-272 bp) at the Pi02 locus. This sub-clonal lineage may have been generated under Egyptian selection pressure conditions and may be considered high-temperature-tolerant, as numerous outbreaks were observed in May when the temperature ranged from 25–35 °C during day and 18–22 °C during night. Nonetheless, it is far from conclusive and needs further elaboration. Tolerance to high temperatures was reported on A1 and A2 lineages in Tunisia [59] with a suggestion that A2 isolates were more tolerant. It was not clear whether the A2 isolates in the Tunisian study were of 13_A2 but they were aggressive and metalaxyl-resistant [59]. All the 71 13_A2 blue lineage isolates tested in this study were fully resistant to metalaxyl, which is consistent with other reports [12,60] and has implications for use of this fungicide in Egypt.

The second cluster in the dendrogram contained 55 isolates of 23_A1, and within this there were ten subclones defined. These data are consistent with other studies that have reported its presence and sub-clonal diversity in southern Europe, Tunisia and Algeria [14,61,62]. In agreement with other work [61], genotype 23_A1 is identical to US-23, and both are reported to be tomato-adapted. Its spread in Europe may have mirrored its spread in the US via trade in tomato plants [61]. It is unclear how this lineage was imported into Egypt, but it may be related to tomato production. It is speculated that the 23-A1 clonal lineage adapted to the Egyptian environmental conditions due to its widespread distribution in both potato and tomato crops.

Lastly, a single isolate (EG-6) that was unrelated to 13_A2 and 23_A1 was detected. It had three novel alleles that were not found in the other two lineages (D13 (118 bp), G11 (204 bp), and SSR6 (242 bp). This genotype has not been reported in other studies or in the large EuroBlight database. The origin of such novel genotypes is unclear. It may be a local recombinant formed from an oospore germination event in an Egyptian crop or represent a genotype imported with potato seed from another part of Europe.

The genome of *P. infestans* consists of an extensive expansion of specific families of secreted disease pathogenicity effector proteins (>700), which are coded in the mobile element of the genome [63]. The pathogen effector, which is a product of an avirulence (Avr) gene, interacts with the corresponding R protein in the plant. The effector proteins can target different sites in infected host plant tissue [64]. These protein–effector interactions are studied to better understand the underlying molecular mechanisms of late blight resistance. All oomycete avirulence genes discovered to date have an RxLR-motif (RxLR=arginine, any amino-acid, leucine, arginine) [44]. The *P. infestans* effectors Avr2, Avr3a, Avr4 and Avr-blb1 are the most studied effectors and belong to the RxLR group [43,44,65].

To understand the underlying mechanisms of how *P. infestans* evades recognition and overcomes resistance genes in potato, the presence/absence of five effector genes that are responsible for virulence or avirulence activity on potato clones carrying R1-R4 genes was screened. In almost every case, the evidence from the effector characterization (Figure 6) matched the results of the differential screening (Figure 3). Blue lineage 13_A2 isolates were virulent on the potato differential carrying R1, and their virulence was associated with the deletion of the Avr1 gene, as reported previously [12]. In contrast, the Avr1 locus was present and amplified in the 23_A1 and misc. isolates. Unexpectedly, despite the presence of avr1, the 23_A1 isolates were also virulent on R1. The avr1 sequences recovered in this study from 23_A1 and the misc. isolate matched other isolates reported to be avirulent [66], so the cause of the virulence is unclear. Perhaps, like with avr2 [42], the gene is not expressed and therefore not recognized by R1. Consistent with studies on other clonal lineages [42], Avr2 was amplified in all six 23_A1 and one misc. isolate tested. No SNPs were detected amongst these isolates resulting in the avirulent form, which was recognized by the R2 plant. Conversely, Avr2 was absent in all eight 13_A2 lineage isolates tested and the 13_A2 isolate was virulent against R2. Unexpectedly, in one 13_A2 isolate (EG-96), a product was amplified. However, it was shown to match the avr2-like gene, indicating a non-specific amplification, which is consistent with other studies and likely a PCR artefact due to primer mismatching [42]. The avr2-like gene was amplified from isolates of all tested samples, and two SNPs were identified that matched other work [42] did not influence the virulence test results. Consistent with other studies [43] Avr3a was present in all tested isolates and contained the homozygous EM or heterozygous EM/KI virulent forms, which corresponded to the virulence test results against R3a in this study. Avr4 was present in all nine tested isolates. Both homozygous and heterozygous forms of Avr4 were virulent, with SNPs matching those previously reported [44]. The findings in the current study widen our understanding of the effector diversity in the 23_A1 lineage. Such data are useful in the long-term understanding of host–pathogen interactions and are critical to the future deployment of durable forms of late blight resistance [65].

## 5. Conclusions

Two clonal lineages of *P*. *infestans* previously reported in Europe were the main cause of potato late blight disease outbreaks in Egypt over the 2009–2012 seasons. Surprisingly, 13_A2 seemed well-adapted to the warm environmental conditions in Egyptian potato crops. Differences in effector profiles were noted between the two lineages with some novel alleles detected. The population of *P*. *infestans* in Egypt is very dynamic and likely to be evolving through local genetic mutations plus migration via trade in potato seed. Further work is needed to update the survey and characterize the clones to exploit the knowledge in potato breeding programs and late blight IPM.

## Figures and Tables

**Figure 1 jof-08-00468-f001:**
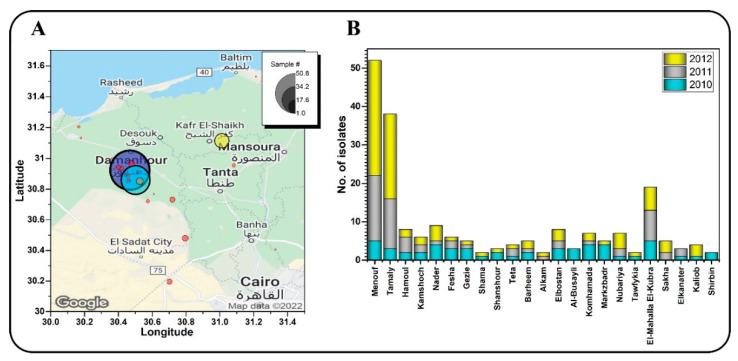
Density map of potato and tomato plants infected with *P. infestans* (**A**), and the number of isolates collected from 23 different locations in 6 Egyptian governorates during three growing seasons (**B**).

**Figure 2 jof-08-00468-f002:**
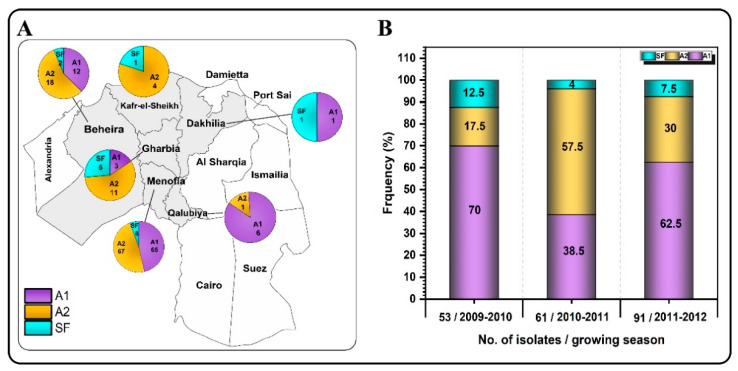
Mating-type frequencies of isolates of *P. infestans* across six Egyptian governorates (**A**) during the 3 years of the study (**B**).

**Figure 3 jof-08-00468-f003:**
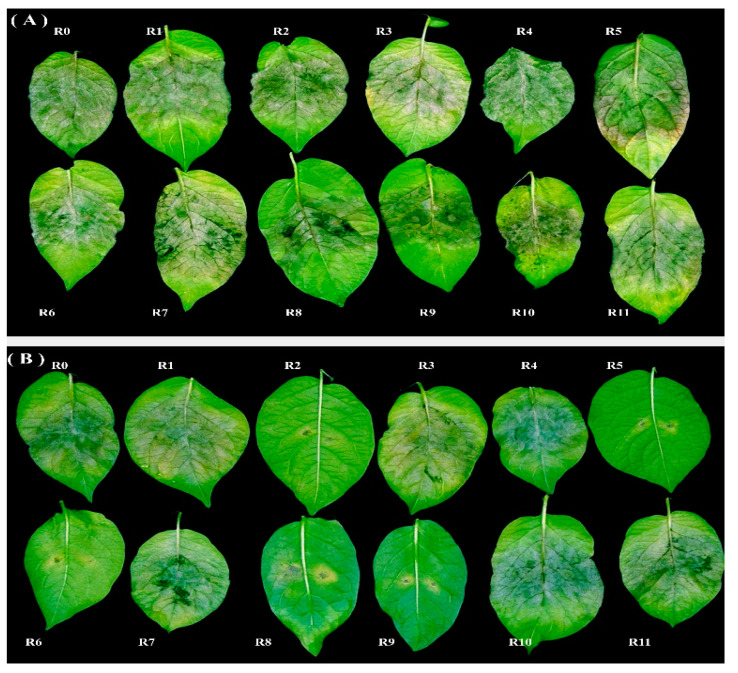
Detached-leaflet assay of two different isolates of *P. infestans,* (**A**) EG-96 and (**B**) EG-75, on 11 differential sets of potato clones carrying R1 to R11, and Craig’s Royal (R0) as a positive control.

**Figure 4 jof-08-00468-f004:**
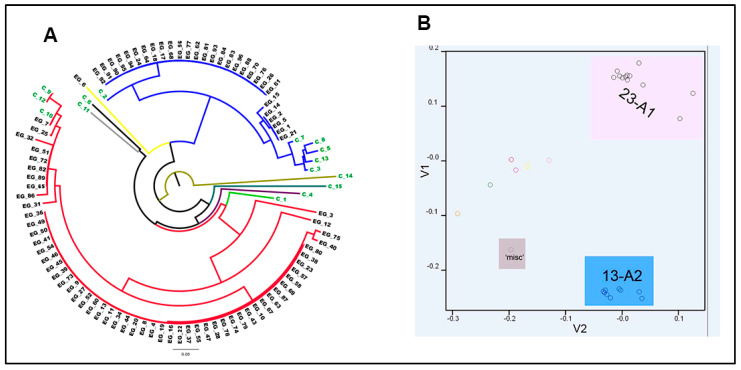
(**A**) Phylogenetic analysis of 85 Egyptian *P. infestans* isolates and 15 reference isolates (green text) based on 12 SSR marker data. The phylogenetic relationship was inferred using the neighbor-joining method examined in POLYSAT. The red lines represent the 23_A1 clonal lineage, the blue line indicates 13_A2 clonal lineage, and the yellow line represents a unique lineage, (**B**) A principal components analysis represents 100 selected isolates that were characterized using 12 SSR markers and examined in POLYSAT. PCA analysis revealed three major clusters: 13_A2 (represented in blue), 23_A1 (pink), and miscellaneous (grey). Reference isolates (without a shaded box) formed independent distributions.

**Figure 5 jof-08-00468-f005:**
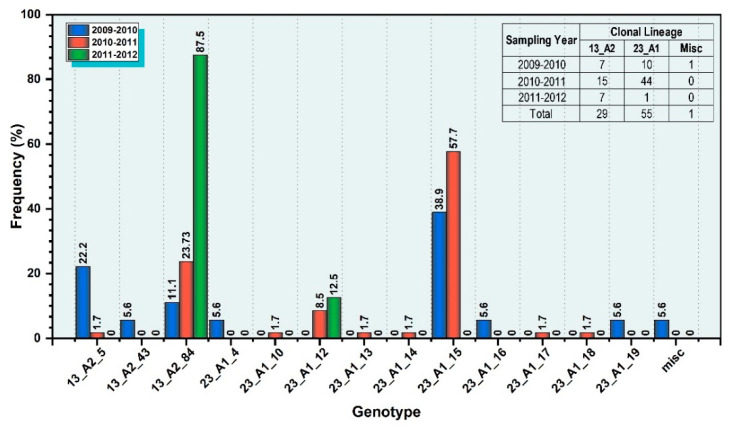
Genotype frequency of the Egyptian populations of *P. infestans* during the 3-year period 2010–2012. Genotype percentage frequency is represented in graph and numbers in table.

**Figure 6 jof-08-00468-f006:**
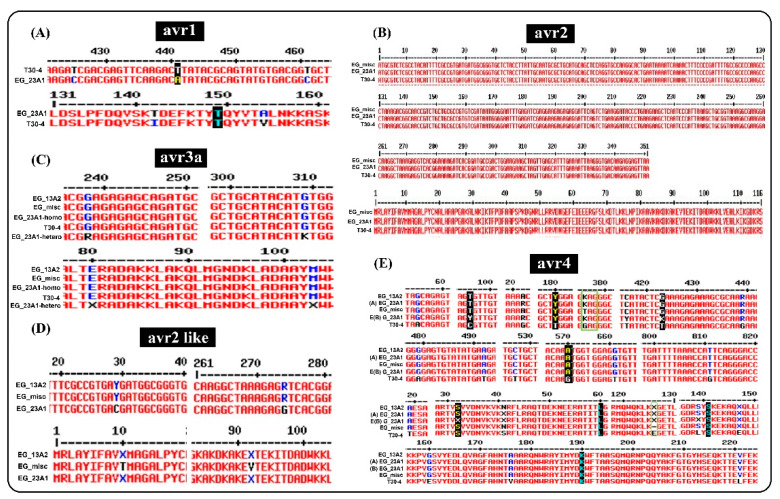
Partial alignments indicating locations of SNPs in the five effector genes of isolates of *P. infestans*; avr1 (**A**), avr2 (**B**), avr3a (**C**), avr2 like, (**D**) and avr4 (**E**). Synonymous mutations within the Egyptian isolates are represented with the black background both for nucleotide and amino acid, while green line box refers to stop codon and truncation at position 126 of AA in the avr4 protein. An X in the amino acid sequence refers to a heterozygous non-synonymous change.

**Table 1 jof-08-00468-t001:** Primers used to detect the presence and absence of effector genes in *P. infestans* isolates.

Primer	Sequence (5′-3′)	References
AVR1F1	TTGCCCTGTTGTGTATGCAT	[12]
AVR1R1	GGCCATCAATTCCTCAGGT	[12]
Pex147F	CCATGCGTCTGGCAATTATGCT	[43]
Pex147R	CTGAAAACTAATATCCAGTGA	[43]
PiAvr4F	ATGCGTTCGCTTCACATTTTGCTGG	[44]
PiAvr4R	CTAAGATATGGGCCGTCTAGCTTGGAG	[44]
PiAVR2_F2	GACCAAACGGCGTACTTCAT	[42]
PiAVR2_R2	CGCCGAGCTCTTAACTCCT	[42]
Piavr2_F7	ACGCTTCTATCCGACAACGA	[42]
Piavr2_R7	ATTGGTGGTAATGCCTGCG	[42]
NitRedF	GGACCGCTGGGCCACTTCAC	[42]
NitRedR	CGCTGGCTTGCAGGCGTACT	[42]

**Table 2 jof-08-00468-t002:** Analysis of mating type, metalaxyl sensitivity, and clonal lineages of *P. infestans* isolates from six governorates of Egypt during three growing seasons.

Sampling Year	Mating Type			Metalaxyl Sensitivity		
	A1	A2	SF	R	S	IR
2010	37	9	7	25	25	3
2011	24	35	2	32	22	7
2012	57	27	7	35	41	15
Total	118	71	16	92	88	25

SF = Self-fertile R = Resistant; S = Sensitive; IR = Intermediate.

**Table 3 jof-08-00468-t003:** Allele sizes identified in the Egyptian population of *P*. *infestans*.

SSR locus	Fragments	Alleles Size
Pi02	4	266, 268, 270, 272
Pi4B	3	205, 213, 217
G11	9	142, 154, 156, 160, 164, 204, 206, 208, 210
Pi04	2	166, 170
Pi63	3	270, 273, 279
Pi70	1	192
D13	8	118, 134, 140, 142, 154, 156, 210, 214
SSR11	2	331, 341
SSR2	2	173, 175
SSR4	3	284, 288, 294
SSR6	3	240, 242, 244
SSR8	2	260, 266

**Table 4 jof-08-00468-t004:** Screening for presence and absence effector genes in the selected *P*. *infestans* Egyptian isolates.

Isolate Name	Genotype	Avr1	Avr2	Avr2-Like	Avr3a	Avr4
EG_02	13_A2_5	-	-	+	+	+
EG_05	13_A2_5	-	-	+	+	+
EG_15	13_A2_43	-	-	+	n/d	n/d
EG_18	13_A2_84	-	-	+	n/d	n/d
EG_90	13_A2_84	-	-	+	n/d	n/d
EG_91	13_A2_84	-	-	+	n/d	n/d
EG_95	13_A2_84	-	-	+	+	+
EG_96	13_A2_84	-	-	+	+	+
EG_11	23_A1	+	+	+	+	+
EG_39	23_A1	n/d	+	+	n/d	n/d
EG_40	23_A1	+	+	+	+	+
EG_75	23_A1	+	+	+	n/d	n/d
EG_86	23_A1	n/d	+	+	n/d	n/d
EG_87	23_A1	n/d	+	+	n/d	n/d
EG_03	23_A1	-	n/d	n/d	+	+
EG_07	23_A1	-	n/d	n/d	n/d	n/d
EG_79	23_A1	+	n/d	n/d	n/d	n/d
EG_06	misc	-	+	+	+	+
T30-4	A1	+	+	-	+	+

Abbreviations or symbols used in the table are denoted as Not determined (n/d), absent (-), and present (+).

**Table 5 jof-08-00468-t005:** The details of the effector genes in the current study with the reference strain T30-40, including the isolate and genotype to which they belong, as well as the accession number of effector genes submitted to GenBank.

No	Effector Gene	Isolate Name	Genotype	Accession Number
1	avr1	EG_40	EU_23_A1_17	MG976587
2	avr1	EG_75	EU_23_A1_18	MG976588
3	avr1	EG_79	EU_23_A1_15	MG976589
4	avr2	EG_06	misc	MG976590
5	Avr2	EG_11	EU_23_A1_15	MG976591
6	Avr2	EG_39	EU_23_A1_15	MG976592
7	Avr2	EG_40	EU_23_A1_17	MG976593
8	Avr2	EG_75	EU_23_A1_18	MG976594
9	Avr2^*^	EG_96	EU_13_A2_84	MG976595
10	avr2-like	EG_05	EU_13_A2_5	MG976596
11	avr2-like	EG_06	misc	MG976597
12	avr2-like	EG_11	EU_23_A1_15	MG976598
13	avr2-like	EG_18	EU_13_A2_84	MG976599
14	avr2-like	EG_40	EU_23_A1_17	MG976600
15	avr2-like	EG_96	EU_13_A2_84	MG976601
16	avr3a	EG_05	EU_13_A2_5	MG976602
17	avr3a	EG_06	misc	MG976603
18	avr3a	EG_11	EU_23_A1_15	MG976604
19	avr3a	EG_40	EU_23_A1_17	MG976605
20	avr4	EG_05	EU_13_A2_5	MG976606
21	avr4	EG_06	misc	MG976607
22	avr4	EG_11	EU_23_A1_15	MG976608
23	avr4	EG_40	EU_23_A1_17	MG976609
24	avr4	EG_96_	EU_13_A2_84	MG976610
25	Avr1	T30-4	A1	XM_002896847
26	Avr2	T30-4	A1	XM_002902939
27	avr3a	T30-4	A1	XM_002898796
28	Avr4	T30-4	A1	XM_002904373

Nomenclature in this table was used [43]. Avr (with a capital A) was used for avirulent form, while avr was used for the virulent form of the gene.

**Table 6 jof-08-00468-t006:** Screening of five RxLR effector genes to identify the presence and type of SNPs in two clonal lineages of *P*. *infestans* isolates.

Effector	Product Size bp	Total SNPs	Syn-SNPs	13_A2	23_A1	No of AA	Source of Allelic Variance
avr1	550	3	1	No ^@^	Yes *	208	Mut or Del
Avr2	500	0	0	No	Yes	116	Del
avr2-Like	480	2	0	Yes	Yes	116	Mut
avr3a	450	2	0	Yes	Yes	147	Mut
avr4	950	13	4	Yes	Yes	287	Mut or Tru

Abbreviation used in the table are synonymous (Syn), amino acid (AA), mutation (Mut), deletion (Del), truncation (Tru), and single nucleotide polymorphism (SNP). It is called N-syn or Syn as the abbreviation. * Gene Present, ^@^ Gene absent.

## Data Availability

All the data related to this study are mentioned in the manuscript and Appendix A.

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
