# Peer review of "Genotypic and Phenotypic Structure of the Population of Phytophthora infestans in Egypt Revealed the Presence of European Genotypes"

_jof, 2022, doi:10.3390/jof8050468_

Round 1
Reviewer 1 Report
P1,L20 - use ‘caused’ one time kindly.
L26 - in the results section it is 57.56, so replace ‘57’ with ‘58’
P3 L11 - replace ‘pre-labeled plastic bags’ with ‘pre-labeled sterilized plastic bags’
P3 L124 - Please add the reference of pea agar media.
P3, L140 - There should be space between number/value and scientific unit e.g., ‘100μg’ replace with ‘100 μg’. Please correct such typo throughout the manuscript.
P4 - Kindly add the PCR mixture recipe at P4.
P4 L165 - kindly add the reference of PCR, who used similar primer or PCR profile.
P4 L169 -Please add the reference of GeneMapper v3.7 (Applied Biosystems)
Table 1 - NiteRedR primer reference is U14405.1. what is it? Please correct it.
P5, L203 - Replace “Of all the collected 205 P. infestans isolates” with “Out of the 205 P. infestans isolates”
In general, I suggest to add/replace 5 references from year 2018-2022 in discussion section.
Kindly make the conclusion simple and expressive.
Reviewer 2 Report
Journal of Fungi
Manuscript Number: jof-1707931
The manuscript entitled "Genotypic and phenotypic structure of the population of Phytophthora infestans in Egypt revealed the presence of European genotypes" by El-Gnainy et al, addresses an interesting and topical issue. The data presented and the methodology applied in the study are appropriate and of good quality. The results were thoroughly analyzed, and discussed adequately. The findings provided very interesting data about the presence of both 13-A2 and 23-A1 lineages in the same region where the chance of sexual recombination can be occured resulted in new lineages. However, there are some re-allignment in the effector genes is needed. After minor revisions, the manuscript can be accepted for publication, as indicated below:
Materials &Methods
- In 2.4 section, Determination of virulence profile: please indicate the growth condition (light, temperature) in which the plants were inoculated and kept in the greenhouse at James Hutton Institute. Please indicate the age of the inoculated whole plants.
- In 2.4 section, please indicate after how many days from the inoculation, the virulence profile was assessed.
- In 2.5 section: the marker PinfSSR3 is not listed in Table S2. Please fix the name of G11 (either G11 as in the M&M) or (PiG11) as it is in Table S2. The marker Pi02 is listed in Table S2 and is not mentioned in M&M.
- In Table S2, please check the fluorescent dyes in PiG11 and in PinfSSR6.
Results
- In “Samples collection and isolation of P. infestans”, please indicate how many isolates were obtained from potato and how many were obtained from tomato. (In Numbers or percentage).
- In: 3.2. Determination of mating type section: in line 205, please add (Figure 2 & Table 2) at the end of the sentence.
- In line 235 modify Figure 3 into Figure 3A
- In line 239 delete 3A and keep only Figure 3B
- In 3.5 section: please match the names of SSR locus in Table 3 with their corresponding names in Table S2.
- In 3.6 section: Egyptian population structure, it was written” Conversely, 23_A1-10, 23_A1-12, 23_A1-13, 23_A1-14, 23_A1-18, and 23_A1-19 were detected only in 2012. Please check these results according to the Figure 5. BUT the genotypes: 23_A1-10, 23_A1-13, 23_A1-14, 23 _A1-17, 23 _A1-18 were detected only in 2011 and 23_A1-19 were detected only in 2010. The genotype 23_A1-12, were detected in both 2011 and 2012.
- In the text sometimes it is written Avr3a and sometimes Avr3 without “a”. please harmonize it in the text and the tables.
- Section 3.3 – Metalaxyl sensitivity assay, better round off the given percentages.
- Table 2, asterisks is mentioned at Metalaxyl sensitivity but not defined in the table footnotes.
- Table 2, re-arrange the footnotes “SF= Self-fertile, R = Resistant; S = Sensitive; IR = Intermediate. Please indicate the abbreviation of
- Table 2, for consistency modify Mis into misc
- Table 2: IR 13_A2 23_A1 Mis should be in bold formats.
- Table 2, although that number of isolates for each clonal lineage was given in the table, but no interpretation of this results was given in the text. Only interpretation was given to the Metalaxyl sensitivity and mating type. Meanwhile it was stated in the discussion section (line 450) that all 13_A2 blue lineage isolates were fully resistant to metalaxyl. The way the data was presented in the table does not reflect the correlation between Metalaxyl sensitivity and clonal lineage.
- In the result section, 3.7. Analysis of effector genes, it is stated that Avr2 was absent in all isolates but present in only one isolate, EG 96, due to mis-amplification of the avr2-like gene. While the data is not shown. So, I prefer to align this Avr2-like with other amplified avr2 like products from different isolates and delete the following sentence
- In line 324, delete the sentence “avr2 was only amplified from one of the eight tested 13_A2 samples (EG-96). Then modify the sentence into avr2 was amplified from all the six 23_A1 and misc isolates tested.
- In Line 357 delete “Only one 13_A2 blue lineage isolate (EG-96) was amplified with these primers but the sequences differed from the other amplified Avr2 gene of the tested isolates.
- In line 359 delete “Twenty SNPs were detected between all tested isolates and the EG-96 isolate and this was consistent with the single 13_A2 product being a mis-amplification of the avr2-like gene as the SNPs were identical to that gene product (data not shown).”
- Table 4, in the column of Avr2 (+) must be turned to (-) with EG-96 as it was avr2-like not avr2.
- In line 356, for consistency please use (misc) as miscellaneous instead of “other”
- In Figure 6D, Avr2 of the reference strain T30-4 must be removed because it is different genes and both form of av2-like should be included in this alignment.
- Figure 5, infestans should be italic.
- Table 6, infestans should be italic.
- Table 5 legends need a revision.
Discussion
- In line 415-416, Does it mean you kept the infected samples since isolation period until the mating type characterization?.
- In line 487: please add “,” after avr1 and before the 23_A1. The sequences recovered
- In line 488: please correct the word isolate to isolates.
- In line 493: in the “Conversely, Avr2 was absent in seven of the eight 13_A2 lineage isolates tested and the 13_A2 isolate was virulent against R2. Unexpectedly, in one 13_A2 isolate (EG-96) a product was amplified. Please update this sentence according to what was requested to modify in the results section above. In more details, modify Avr2 was absent in 8 isolates and delete “Unexpectedly, in one 13_A2 isolate (EG-96) a product was amplified”.
After addressing these edits/modifications I think the paper can be accepted for publication.
